# ON THE ENTROPY OF LANGUAGE MODELS IN GETTING SEMANTIC FROM TOKENS

## ABSTRACT

Large language models (LLMs) are widely recognized for their exceptional capacity to capture semantic meaning. Yet, there remains no established metric to quantify this capability. In this work, we introduce a quantitative metric, **Information Emergence (IE)**, designed to measure LLMs' ability to extract semantics from input tokens. We formalize "semantics" as the meaningful information abstracted from a sequence of tokens and, leveraging information theory, quantify this through comparing the reduction in entropy observed for a sequence of tokens (macro-level) and individual tokens (micro-level). To achieve this, we design a light-weight estimator to compute the mutual information at both micro and macro levels for each transformer layer, which is agnostic to different tasks and language model architectures. We apply IE in both synthetic in-context learning (ICL) scenarios and natural sentence contexts. Experiments show a high-level informativeness of our metric reflected in semantic faithfulness, sensitivity and connection with emergence. In addition, we highlight some interesting findings: 1) IE explains why ICL offers clearer semantics and benefits compared to natural text through changes in entropy. 2) We could associate certain hallucination phenomenon with increased variance in IE. 3) IE can effectively differentiate between human-written and LLM-generated text, proving especially useful for extremely large and closed-source language models. Our codes are available at: `https://anonymous.4open.science/r/Emergence/`.

## 1 INTRODUCTION

One of the most elusive and captivating attributes of large language models (LLMs) is their ability to learn semantics from inputs across diverse domains(Chen, 2023; Chang et al., 2024; Minaee et al., 2024), a feature that owes much to a cross-pollination of unsupervised training and next-token prediction (NTP) mechanisms. It has stimulated numerous significant research directions, such as in-context learning (ICL) (Min et al., 2022a;b; Wies et al., 2024; Ye et al., 2023; Kossen et al., 2023; Swaminathan et al., 2024), emergence capabilities (Wei et al., 2022; Schaeffer et al., 2023; Srivastava et al., 2023; Lu et al., 2023; Yu & Dong, 2022; Liu et al., 2024), and hallucination investigations (Rawte et al., 2023; Ji et al., 2023; Zhang et al., 2023; Deemter, 2024).

However, the capability of LLMs to capture semantics from texts is challenging to quantify and thereby, to evaluate. Numerous existing tasks indirectly reflect similar capabilities through evaluating LLMs' performances (e.g., accuracy) on a specific task, such as "instruction following" (Zeng et al., 2023), "searching" (Sun et al., 2023), and "reasoning" (Yang et al., 2024). Nevertheless, these evaluation methods rely on manually curating datasets and tasks tailoring different aspects, resulting in time-consuming and domain-specific findings. In addition, these evaluations typically focus on coarse-grained text, not providing interpretations for the behavior of finer-grained tokens. Lastly, existing evaluation metrics which vary across different tasks can lead to varied performances, and even contradicting conclusions (Schaeffer et al., 2023).

In response to the above limitations, we propose a task-agnostic and closed-form metric, which we refer to as **Information Emergence (IE)**[1], designed to reflect and deterministically quantify the

---

[1]The "Information Emergence" here and "Emergence" in LLM-related research are two different notions, we discuss their difference in Section 2.1 and Section 4.3.

ability of LLMs to extract meaningful semantics from input tokens. To begin with, we construct a mathematical formalism capable of modeling semantics. In essence, semantics naturally emerge as a meaningfully organized ensemble of tokens (Hilpert & Saavedra, 2020; Apidianaki, 2023). Consequently, tokens are considered microscopic (micro) observations with sophisticated patterns in a sentence, whereas semantics represent macroscopic (macro) observations emerging with more predictable behaviors. Inspired by information theory, we formalize the model's proficiency in semantic understanding, i.e., information emergence, as the difference of the entropy reduction between micro-level and macro-level. In another word, **a better model proficient in deriving semantics from tokens, in comparison to other models, ought to render a higher entropy reduction for a global sequence than for a single token**.

To compute IE in transformer models, as discussed earlier, we need to mathematically measure entropy reduction for both micro and macro levels. Given the auto-regressive nature of the NTP mechanism, at any layer $l$ in transformer, the most micro-level transition can be naturally framed as the probability $p_{h_l^0|h_{l-1}^0}$ for an isolated token $h_l^0$, whereas the macro-level transition can be formulated as $p_{h_l^T|h_{l-1}^0, h_{l-1}^1, ..., h_{l-1}^T}$ across $T$ tokens. We resort to the mutual information between successive transformer layers and adopt a practically effective estimation algorithm motivated by (Belghazi et al., 2018) which is suitable for high-dimensional continuous representations. Therefore, we can measure the IE value for any token at any transformer layer, reflecting the strength of the LLM's capability in extracting semantics from the historical context.

We devise a suite of comprehensive experiments encompassing two different scenarios. In the first scenario, we curate a group of synthetic datasets under the ICL setting with different context domains. In the second scenario, we collect two wild datasets consisting of real-world natural language questions/answers. Under both scenarios, we experiment with different LMs including GPT-2 (Radford et al., 2019), GEMMA (Team et al., 2024), and OpenLlama (Computer, 2023). In alignment with our hypothesis, we show that IE offers a high-level informativeness through semantic faithfulness and sensitivity - the richer the semantics, the higher the IE. Additionally, a set of experiments conducted across model sizes have indicated the potential association between IE and Emergence. Furthermore, we obtain 3 interesting findings:

1. IE increases token-by-token in natural texts, whereas, in ICL-style texts, IE increases only when a new demonstration appears.

2. There is a strong correlation between specific hallucination phenomenons and a high variance in IE scores.

3. Distinctive patterns in IE have been observed between human-written and LLM-generated texts, revealing IE's potential in automatically recognising LLM generations.

Overall, the main contributions could be summarized below:

- We introduce IE, a novel, reasonably validated, and task-agnostic metric to deterministically quantify the semantic understanding capability of LLMs.

- We introduce a light-weight implementation method for evaluating IE, which can be applied to extremely large and closed-source LMs like GPT-4 (Achiam et al., 2023).

- Empirical evidence demonstrates that IE can uncover previously unknown and essential patterns in areas such as ICL, Emergence, and hallucination.

## 2 RELATED WORK

### 2.1 INFORMATION EMERGENCE AND EMERGENCE

Emergence is defined as a capability that does not exist in smaller models but appears in larger ones (Srivastava et al., 2023; Lu et al., 2023; Yu & Dong, 2022; Liu et al., 2024). Most commonly, as the model size increases, the performance on many tasks rapidly improves. Several studies have posited explanations for Emergence: the qualitative change resulting from quantitative behavior (Wei et al., 2022), the random combination of linguistic skills (Arora & Goyal, 2022), and principles that can be extrapolated from smaller models (Schaeffer et al., 2023).

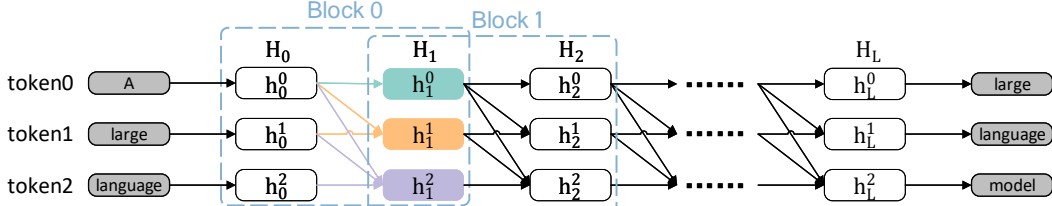

Figure 1: The analogy of auto-regressive process in NTP to Markov process. Taking the output representation of token2 in Block 0 ($h_1^2$) as an example, which receives information from input representations of $h_0^0$, $h_0^1$, and $h_0^2$, satisfying $p_{h_{l+1}^2 | H_l^{\leqslant t}} = p_{h_{l+1}^2 | h_l^0, h_l^1, h_l^2}$.

IE is a concept defined and validated in Information Theory (Bedau, 1997; 2008). It describes phenomena observable at the macroscopic level but unobservable at the microscopic level. Nevertheless, only limited empirical experiments are conducted to reflect a similar pattern of abrupt improvement between IE and Emergence with increased model size.

## 2.2 EVALUATION ON LLM CAPABILITIES

The prevalent body of research extensively measures the capabilities of LLMs across various tasks by employing substantial benchmark datasets (Srivastava et al., 2023; Wang et al., 2024; Zhu et al., 2024). Additionally, a significant amount of research focuses on the performance of LLMs concerning specific capabilities such as adaptability to different domains (Afzal et al., 2024), human-like cognition (opinions, attitudes, etc.) (Ma et al., 2024), followed with input instructions (Zeng et al., 2023), text searching capability (Sun et al., 2023), and reasoning ability (Yang et al., 2024). Moreover, numerous studies have also considered the performance of LLMs in collaborative evaluation with humans (Kim et al., 2024; Zheng et al., 2023). In contrast to these studies, our work concentrates on an essential yet abstract ability of LLMs - the ability to extract semantics from tokens.

## 3 METHOD

### 3.1 HOW TO MODEL SEMANTIC IN LLMS

In this paper[2], we identify the transformer block as the fundamental unit. Specifically, we employ $l = 0, 1, \ldots, L - 1$ to index transformer blocks within a language model, where $L$ represents the total number of blocks. For instance, GPT-2 XL (1.6B parameters) comprises 12 blocks ($L = 12$), and Gemma-2B totals 18 blocks ($L = 18$). For any transformer block $l$, given an input sequence of token length $T$ and hidden state dimension $D$, the input representation is given by $H_l = \{h_l^0, h_l^1, h_l^2, \ldots, h_l^{T-1}\}$ and the output representation is $H_{l+1} = \{h_{l+1}^0, h_{l+1}^1, h_{l+1}^2, \ldots, h_{l+1}^{T-1}\}$, where $H \in \mathbb{R}^{T \times D}$ and $h^t \in \mathbb{R}^{1 \times D}$. Without loss of generality, we hypothesize that the multi-layer blocks constitutes a Markov process.

**Hypothesis 1** (Markov Process Analogy). *The auto-regressive process of NTP mechanism in multi-layer blocks undergoes a Markovian stochastic process following a transition probability of any $h_{l+1}^t$ with $p_{h_{l+1}^t | h_l^0, h_l^1, h_l^2, \ldots, h_l^t}$, simply denoted by $p_{h_{l+1}^t | H_l^{\leqslant t}}$.*

To simplify the analogy for easier understanding, Figure 1 omits normalization layers, MLP layers, and residual structures between transformer blocks, and thus the output of $l$-th block is directly considered as the input to $l + 1$-th block ($H_{l+1}$). However, in all our real implementations, we retain the exact transformer output at every layer, i.e., $h_{l+1} = h_l + attention(h_l) + MLP(h_l)$.

Accordingly, we could categorize token variables within each sequence into two distinct categories: **microscopic (micro) variables** and **macroscopic (macro) variables**. A micro variable refers to a

---

[2]Please note that this paper addresses decoder-only autoregressive language models, while other language models that do not meet the requirements, such as BERT (Devlin et al., 2018) family, are not considered within the scope.

token which is solely influenced by a single token as the input. For instance, $h^0$ satisfies $p_{h_{l+1}^0|h_l^0}$. Whereas macro variables aggregate information from all micro variables and thus encompass tokens which are influenced by all the tokens within the sequence as the input. An example could be $h_{l+1}^{T-1}$ which satisfies $p_{h_{l+1}^{T-1}|h_l^0,h_l^1,\ldots,h_l^{T-1}}$.

In summary, the NTP mechanism can be viewed as a behavior that increasingly coarsens from the most micro to the most macro scale and finally forms meaningful semantics. Hence, the macro level represents the semantics level and the micro level indicates the token level. Our definition of IE, hence, represents the phenomenon observable at a semantics level, yet unobservable at a token level during a dynamic process (instantiated in Example 1).

**Example 1.** *Given $T$ binary tokens $H_l = \{h_l^0, h_l^1, \ldots, h_l^{T-1}\} \in \{0,1\}^T$ as inputs, for simplicity, we assume all variables are micro variables: $\forall h_{l+1}^t \in H_{l+1}$ satisfies $p_{h_{l+1}^t|h_l^t}$ (the simplest Markov process, and in the subsequent part of this Example, we use $p$ to simply denote this transition probability). The output representations are also binary, i.e., $H_{l+1} \in \{0,1\}^T$. We assume an evolution rule which enables the parity of the sum of all output variables equal to the sum of all inputs with probability $\gamma$. If $H_l$ satisfies the uniform distribution, the evolution rule entails the probability of the output $H_{l+1}$ :*

$$p(H_{l+1}|H_l) = \begin{cases} \frac{\gamma}{2^{T-1}}, \ if \ \biguplus_{t=0}^{T-1} h_{l+1}^t = \biguplus_{t=0}^{T-1} h_l^t \\ \frac{1-\gamma}{2^{T-1}}, \ otherwise \end{cases} \tag{1}$$

*where $\biguplus_{t=0}^{T-1} h_l^t := 1$ if $\sum_{t=0}^{T-1} h_l^t$ is even and $\biguplus_{t=0}^{T-1} h_l^t := 0$ if odd. For example, if $H_l = \{0,0,0\}$, $H_{l+1}$ can be one of $\{0,0,0\}, \{0,1,1\}, \{1,0,1\}, \{1,1,0\}$ with probability $\gamma$, leading to $\frac{\gamma}{2^{3-1}}$ chance for each candidate above. Each of the remaining value for $H_{l+1}$ has probability $\frac{1-\gamma}{2^{3-1}}$.*

*With the assumption of micro dependency $p_{h_{l+1}^t|h_l^t}$, we can derive the distribution of a **micro variable** as $p(h_{l+1}^t=0|h_l^t) = p(h_{l+1}^t=1|h_l^t) = 0.5$. Finally, let $h^{ma}$ be the macro variable with $h^{ma} = \biguplus_{t=0}^{T-1} h^t$. Then the distribution of the **macro variable** becomes:*

$$p(h_{l+1}^{ma}|h_l^{ma}) = \begin{cases} \gamma, \ when \ h_{l+1}^{ma} = h_l^{ma} \\ 1-\gamma, \ when \ h_{l+1}^{ma} \neq h_l^{ma} \end{cases} \tag{2}$$

Example 1 elucidates an interesting phenomenon of IE: The macro variable $h^{ma}$ is not induced by any individual micro variable but a collective of them. As a result, it shows different phenomenon from any micro variable. Moreover, we note that the same evolution rule applies to both micro and macro variables. This is known as **Supervenience Hypothesis** (Bedau, 1997; 2008):

**Hypothesis 2** (Supervenience)**.** *When the properties of the micro-level mechanisms of a system are fixed, so are the properties of all its macro levels.*

Hypothesis 2 explicates that the "new phenomenon" in macro systems does not materialize *ex nihilo*. The micro level is causally and mechanically complete, and there is no room left for any causal and mechanical contribution at the macro level. In other words, the semantics that emerged in macro has irreducible causal and mechanical power in practice but not in principle, just because the token perspective is too fine-grained to observe such a phenomenon.

Given the Supervenience hypothesis and motivated by more endeavors about information theory (Rosas et al., 2020; Hoel et al., 2013; 2016), we can define IE in LLMs as:

**Definition 1** (Information Emergence in LLMs)**.** *For any transformer block $l$, let $h_l^{ma}$ be the **macro variable** ($h_l^{ma}$ satisfies $p_{h_{l+1}^{ma}|H^{\leqslant T}}$) and let $h_l^{mi}$ be the **micro variable** ($h_l^{mi}$ satisfies $p_{h_{l+1}^{mi}|h_l^{mi}}$), $MI(\cdot, \cdot)$ represents the mutual information, thus the strength of IE in block $l$ can be described as:*

$$E(l) = MI(h_{l+1}^{ma}, h_l^{ma}) - \frac{1}{T} \sum_{t=0}^{T-1} MI(h_{l+1}^{mi\_t}, h_l^{mi\_t}) \tag{3}$$

Definition 1 describes how to estimate the IE metric. To illustrate, suppose an input sequence contains three tokens *'large language model'*, with their representations at the $l$th block denoted as $H_l = \{h_l^0, h_l^1, h_l^2\}$. To compute the mutual information at the micro level, we need to make

sure each micro token is positioned at the beginning of the sequence to avoid the influence from other tokens due to the auto-regressive nature of the NTP mechanism. Specifically, we obtain the first micro variable $\{h_l^{mi\text{-}0}\}_{l=0}^{L-1}$ where $h^{mi\text{-}0} = h^0$ corresponds to 'large' derived by going through the transformer model for the input 'large language model'. The second micro variable $\{h_l^{mi\text{-}1}\}_{l=0}^{L-1}$ corresponds to 'language' given by going through the transformer model one more time for the modified input 'language model'. The third micro variable proceeds in a similar manner by removing the first two tokens in the original input. These modified inputs ensure that each micro variable only depends on itself in the previous block. Meanwhile, the macro variable $\{h_l^{ma}\}_{l=0}^{L-1}$ is given by $h_l^{ma} = h_l^2$ for the original input sequence 'large language model'. Finally, we have $E(l) = MI(h_{l+1}^{ma}, h_l^{ma}) - \frac{1}{3}(MI(h_{l+1}^{mi\text{-}0}, h_l^{mi\text{-}0}) + MI(h_{l+1}^{mi\text{-}1}, h_l^{mi\text{-}1}) + MI(h_{l+1}^{mi\text{-}2}, h_l^{mi\text{-}2}))$.

Notably, We measure IE by comparing the macro changes with the micro changes across blocks, instead of directly computing $MI(h_l^{ma}, h_l^{mi})$ at the same block, justified by the Supervenience hypothesis which posits that under the same state, micro and macro variables do not exhibit emergence phenomenon due to the completeness of the micro level. We use Example 2 to explain why the mutual information between macro variables and micro variables is different:

**Example 2.** *Continuing from Example 1, for simplicity, we assume $\gamma$=1, $T$=3. Assuming block $l$ undergoes a uniform distribution, we have $p(h_l^{ma}$=0$) = p(h_l^{ma}$=1$) = 0.5$ and $p(h_l^{mi\text{-}t}$=0$) = p(h_l^{mi\text{-}t}$=1$) = 0.5$. Without loss of generality, let's assume $h_l^{ma} = 0$ and $h_l^{mi\text{-}t} = 0$. According to Example 1, we have $p(h_{l+1}^{mi\text{-}t}$=0$|h_l^{mi\text{-}t}) = p(h_{l+1}^{mi\text{-}t}$=1$|h_l^{mi\text{-}t}) = 0.5$. On the contrary, $p(h_{l+1}^{ma} = 0|h_l^{ma}) = \gamma = 1$. Hence, the IE value becomes $E(l) = 1 - \frac{1}{3}(0 + 0 + 0) = 1$.*

Example 2 elucidates that the difference in mutual information stems from a coarse-graining that transitions from individual elements (micro) to an aggregate whole (macro). From an information theory perspective, $E(l) > 0$ indicates that when the function of transformer block $l$ results in a higher reduction of uncertainty (entropy) on the whole sequence (macro variable) compared to the individual tokens (micro variables), there is a higher chance of capturing the collective semantics. Consequently, IE can be briefly understood as **"how confidence with which a language model, based on previous tokens, definitely predicts the next token with a lower entropy in semantics"**.

### 3.2 How to Estimate IE in LLMs?

It is not feasible to directly compute the mutual information in Eq. 3 using Kullback-Leibler (KL) divergence, as the input lies in a high-dimensional continuous space. To address that, we resort to an approximation method using mean values proposed in Belghazi et al. (2018):

$$D_{KL}(\mathbb{P}||\mathbb{Q}) = \limsup_{f:\Omega\to\mathbb{R}} E_{\mathbb{P}}[f] - log(E_{\mathbb{Q}}[e^f]) \qquad (4)$$

where $f$ represents a function that maps $\mathbb{Q}$ to Gibbs distribution by $d\mathbb{G} = \frac{1}{Z}e^f d\mathbb{Q}$, where $Z = E_{\mathbb{Q}}[e^f]$. Naturally, $f$ can be a neural model. Thereby, Equation 4 can be equivalently represented as optimizing the error function $\mathcal{L}$:

$$\mathcal{L} = \frac{1}{B}\sum_{b=1}^{B}(f_\theta(x^b||y^b)) - \log(\frac{1}{B}\sum_{b=1}^{B}e^{f_\theta(x^b||y^{i\neq b})}) \qquad (5)$$

where $\theta$ denotes the parameters for $f$, $||$ denotes the concatenation operation and $B$ is the batch size. $x, y \in \mathbb{R}^D$ are two inputs for computing the mutual information $MI(x, y)$. $x^b||y^b$ corresponds to sampling from the joint distribution $P_{XY}$, while $x^b||y^{i\neq b}$ corresponds to sampling from the marginal distribution $P_X$ and $P_Y$[3]. When $\mathcal{L}$ converges to the minimum $\hat{\mathcal{L}}$, we can obtain the final estimated mutual information as $MI(x,y)=-\log_2^e*\hat{\mathcal{L}}$. (More details and proofs are shown in Belghazi et al. (2018).)

To get the IE value $E(l)$ in Eq. 3, we compute $MI(h_{l+1}^{ma}, h_l^{ma})$ by replacing $x^b$ and $y^b$ in Eq. 5 with $h_{l+1,s}^{ma}$ and $h_{l,s}^{ma}$ obtained by applying the LM to the same input sequence $s$, whereas replacing $y^{i\neq b}$ with $h_{l,s'}^{ma}$ using a different sequence $s' \neq s$. Similar operations apply when computing $MI(h_{l+1}^{mi\text{-}t}, h_l^{mi\text{-}t})$. (Refer to Appendix A for a complete algorithm for estimator.)

---

[3]In our implementation, the batch size was increased to encompass the entirety of the sample set to ensure the rationality of $P_{XY}$, $P_X$ and $P_Y$.

### 3.3 IMPLEMENTATION

Our algorithm requires that the number of samples is sufficiently large (over 300k) to provide a good estimate of the mutual information. Meanwhile, the length of each sequence within a dataset should be kept the same to facilitate position-wise observation and meaningful computation. Due to resource constraints, our comparative analysis is limited to GPT2-large (812M), GPT2-XL (1.61B), GEMMA (2.51B), and OpenLlama (3B) models. Fortunately, this parameter range is sufficient to observe variations and regularities of IE. For those LLMs with extremely large size or closed resource (e.g., GPT-4, Claude3, etc.), we design another efficient strategy that enables their IE evaluations as shown in Section 5.3. All computational experiments can be conducted on **one** NVIDIA GeForce RTX 3090 GPU[4]. The estimator $f$ in Eq. 4 is a model of a 10-layer neural network comprising linear layers and leaky ReLU activation functions, where each linear layer's output dimension was half of its input dimension. We set the batch size to 300,000 to ensure the stability of the sampling distribution, thereby guaranteeing robust results. Additionally, the learning rate was initially set at 1e-4 and was polynomially decayed to 1e-8 within 10k epochs. We examine the IE value of LLMs under two distinct settings: **ICL** with few-shot examples and **natural sentences** without demonstrations.

#### 3.3.1 ICL SCENARIO

Since existing datasets (e.g., SST-2 (Socher et al., 2013), AGNews (Zhang et al., 2015), and EmoC (Chatterjee et al., 2019)) do not meet the requirement of the same sequence length, we synthesized a set of simple few-shot samples having token length and positions aligned across different sequences. We curate three different datasets encompassing three different domains, each containing sequences of few-shot single-token entities with commas:

- **Country**: We select 25 countries from the Vocabulary as *entities*, each represented by 1 token (e.g., *'Canada'*, *'Russia'*). Each shot consists of one *entity* followed by a *comma*, totaling 2 tokens. We constructed $25 * 24 * 23 * 22 = 303,600$[5] input sequences, each comprising 8 tokens (4 different shots), such as *"France, Mexico, Egypt, Russia,"*.

- **Animal**: Similarly, we select 16 animals as *entities*, and construct $16 * 15 * 14 * 13 * 12 = 524160$ input sequences comprising 10 tokens (5 different shots), such as *"Fox, Pig, Penguin, Rabbit, Cock,"*.

- **Color**: We select 15 colors as *entities*, and construct 360360 samples comprising 10 tokens (5 different shots), such as *"red, orange, yellow, green, blue,"*.

In the experiment, we observe that each *entity*, treated as a micro variable (i.e., the first token), produces similar mutual information across different positions. Consequently, in this section, we only use the *entity* in the first position to compute the mutual information to approximate the mean of all micro variables. Moreover, $E(l)$ also acts analogously in each block, so we utilize the mean of $\{E(l)\}_{l=0}^{L-1}$ to show the IE. By varying the position $t$ of the macro variable (last token in a sequence), we use the following equation to compute the IE at length $t$:

$$\widehat{E}(t) = \frac{1}{L}\sum_{l=0}^{L-1} E(l)_{ma=t-1} = \frac{1}{L}\sum_{l=0}^{L-1}(MI(h_{l+1}^{t-1}, h_l^{t-1}) - MI(h_{l+1}^0, h_l^0)) \tag{6}$$

#### 3.3.2 NATURAL SENTENCE SCENARIO

We randomly select 300,000 natural sequences, each consisting of 8 tokens, from OpenOrca (Lian et al., 2023) and OpenHermes (Teknium, 2023), respectively. OpenOrca and OpenHermes are both large-scale, multi-domain QA datasets. These sequences were selected to ensure that the first token in each sequence is the actual beginning of a sentence.

In our experiments, we observe potential discrepancies in mutual information for individual tokens at different positions within a sentence (i.e., micro mechanisms in different positions are not consistent).

---

[4]For easy deployment, we split the representations of LLMs into several 768-dimension segments and the final representation is the mean of these segments.

[5]The number of shots is decided to ensure the number of combinations in each category is over 300000.

These discrepancies are detailed in Appendix C). Consequently, for the mutual information of micro-level variables, we adhere to Equation 3 which averages the micro MI at each position:

$$\widehat{E}(t) = \frac{1}{L} \sum_{l=0}^{L-1} E(l)_{ma=t-1} = \frac{1}{L} \sum_{l=0}^{L-1} (MI(h_{l+1}^{t-1}, h_l^{t-1}) - \frac{1}{t} \sum_{m=0}^{t-1} MI(h_{l+1}^m, h_l^m)) \tag{7}$$

## 4 INFORMATIVENESS

### 4.1 SEMANTIC FAITHFULNESS

We have designed several experiments to substantiate the **semantic faithfulness** of IE. We observed the change in IE and other popular metrics (Exact Match, Accuracy, and model loss) with the increase in the number of tokens, using the samples from the OpenOrca dataset. Figure 2 demonstrates the change in their values, with increasement $> 0$ representing the value increasing from that in the previous token. Only IE consistently exhibits an upward trend (i.e., $> 0$), which aligns with the intuition: what a sentence intends to convey is increasingly deterministic along with the increasing number of tokens. Moreover, the low variance (reflected as the shaded area in Figure 2) in IE values exhibits commendable stability compared to other metrics.

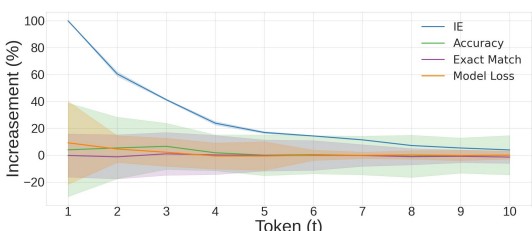

Figure 2: The increasement of IE, EM, Accuracy, and model loss for GPT2-XL in comparison to the previous token: increasement $= (value(t) - value(t-1))/value(t)$, where $value(t)$ represents the value at token $t$. Therefore, a positive increasement $(> 0)$ indicates an increase in the metric value, and a decrease vice versa.

### 4.2 SEMANTIC SENSITIVITY

Subsequently, we aim to examine the **semantic sensitivity** of IE, particularly its ability to reflect differences when minor perturbations are introduced into the semantics. Consequently, we conducted a series of ablation studies to modulate certain factors (such as dataset size, attributes, tasks, and format) individually. We treat the performance of GPT2-XL on the "country" dataset as a baseline. Appendix D details the variations when different factors are changed. It was observed that IE increases with the model's size. This corroborates the rationale that a model with a larger size generally has better capability to determine semantics. In addition, our study also identifies variations in IE against different tasks and prompts, which also resonates with findings from prior research (Lu et al., 2023; Yu & Dong, 2022; Liu et al., 2024).

### 4.3 CONNECTION TO EMERGENCE

Moreover, We demonstrate that IE manifests a steep ascend within the parameter range of $10^8$ to $10^{10}$ across 8 arithmetic tasks, which is detailed in Appendix B). Given the confines of computational resources, we were able to select 8 models within the parameter range of GPT2 ($1 * 10^8$), GPT2-large ($7 * 10^8$), GPT2-XL ($1 * 10^9$), Gemma ($2 * 10^9$), OpenLlama ($3 * 10^9$), GPT-J ($6 * 10^9$), Gemma ($7 * 10^9$), and GPT-NeoX ($2 * 10^{10}$). In light of the existing evaluation work, Big-Bench (Srivastava et al., 2023), we discovered the emergent phenomena within the arithmetic tasks emerge within the parameters of $[10^8, 10^{10}]$. Consequently, the association between the performance and IE values of 8 arithmetic tasks was investigated, as shown in Figure 3. For model performance, we directly adopt the default settings of the Big-Bench benchmark. As for IE, we took the average of the IE values of the initial five output tokens to be the final result.

A marked enhancement in task performance occurs once effective parameters reach $10^{10}$, thereby showcasing an emergent phenomenon. The average IE experiences a substantial surge within the same parameter range ($10^9$ to $10^{10}$). As a pioneering work proposing a quantitative metric to reflect the level of semantics deterministically, we believe our method could also greatly benefit further research on Emergence.

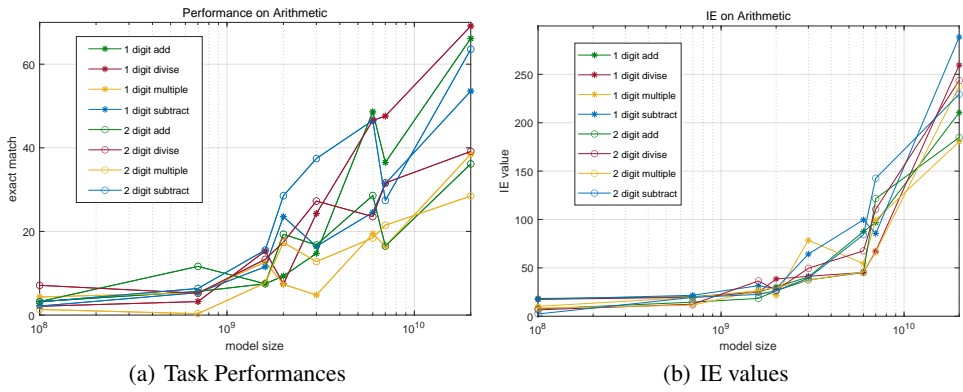

(a) Task Performances         (b) IE values

Figure 3: IE and Model Performance with model size increasing in Arithmetic

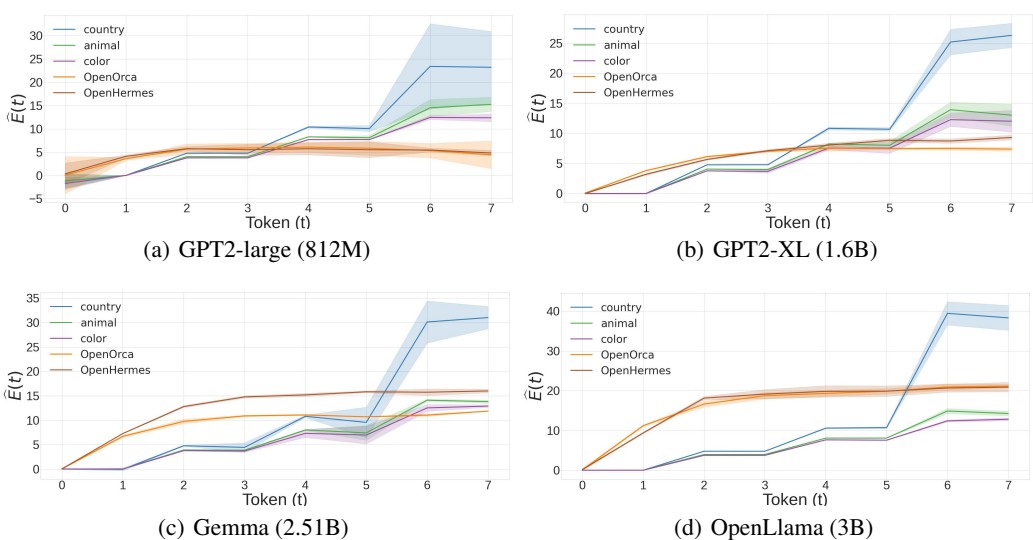

(a) GPT2-large (812M)        (b) GPT2-XL (1.6B)

(c) Gemma (2.51B)        (d) OpenLlama (3B)

Figure 4: $\widehat{E}(t)$ on ICL and natural scenarios with mean and variance.

## 5 FINDINGS

### 5.1 IE INCREASES ONLY WHEN A NEW DEMONSTRATION APPEARS

Figure 4 illustrates that IE naturally becomes higher with increased tokens. However, there is a strikingly different trend between ICL and natural scenarios (containing natural sentences). In a natural scenario, IE increases with each successive token and achieves a rapid convergence (around the 6th token), whereas, under the ICL scenario, IE only increases when a new demonstration emerges (at positions of the 2nd token, 4th token, 6th token) [6], but with a higher upper bound and requiring more tokens to reach to the highest value.

We subsequently investigate how many demonstrations are needed before IE ceases to increase. Table 4 indicates that the three ICL categories under study tend to saturate at the 7th demonstration (though this does not suggests a generic ICL phenomenon). Moreover, we test whether increasing the number of tokens within each demonstration would maintain this "stepwise elevating" pattern. Figure 6 shows that the IE scores within each demonstration does not change when the length of each demonstration is increased to 5 tokens, 6 tokens, and 7 tokens.

---

[6]The complete record of every token's mutual information is detailed in Table 5, showing the example of GPT2-XL on the Animal category.

Hence, we can interpret ICL's role in enhancing semantic determinability: ICL bolsters semantic determinability via demonstrations, where increasing the number of demonstrations can increase the ability of capturing semantics beyond natural text, but eventually saturates after a certain quantity.

Concurrently, disparate performances observed across the two datasets and three model families suggest that the domain of the training data and preprocessing methodologies are likely critical factors, as further supported by the evaluations of individual tokens at different sentence positions elaborated in Appendix C. For instance, GPT2-large and GPT2-XL, which share the same dataset (40GB WebText), preprocessing methods, and model architecture, exhibit a common characteristic: the mutual information of the first token is consistently lower than the average mutual information of micro-level variables.

## 5.2 Higher IE with Large Standard Deviation Corresponds to Certain Hallucination Behaviors

Figure 4 indicates that IE becomes unstable as the number of demonstrations increases. To further study this observation, we explicitly report the IE and standard deviation (s.d.) in Table 1 and compute the accuracy of the generations[7] spanning over different numbers of shots. As can be seen from rows 1-9 of Table 1, as IE ceases to grow and the s.d. reaches the peak, the LM displays a higher probability of generating inaccurate responses. From a closer look, we discover that oftentimes, the LM fails to generate new entities due to "error repetition" (explanations and some examples can be found in Appendix E.3). This is aligned with existing study (Zhang et al., 2023) related to hallucination. Specifically, LLMs struggle to correct themselves after generating an erroneous output, consequently resulting in stagnation and fluctuation in IE value.

However, this should not be confused with the power of ICL in exploiting more complex patterns effectively with more "shots" as the input. Different from the above observation, an increasing number of shots tend to bring higher accuracies under more complex scenarios. As shown in rows 10-15 of Table 1, we design four challenging tasks: 'Asia' and 'Europe' only provide countries in Asia and Europe, respectively, as input demonstrations; 'Size' contains animals arranged by size from smaller to larger; 'Alphabet' sorts the entities alphabetically based on the first letter. The accuracy results indicate that LLMs require more shots to capture complex patterns compared to simple patterns. Thus, it prompts us to conjecture if the stagnation and fluctuations in the IE are associated with another hallucination: with excessive shots, LLMs may perceive more complex patterns beyond the surface (or even actual) appearance. In short, the correlation between IE s.d. and hallucination would offer novel insights into the future development of hallucination detection and mitigation.

| IE value by each shot | | | | | | | |
|---|---|---|---|---|---|---|---|
| Statistics | shot1 | shot2 | shot3 | shot4 | shot5 | shot6 | shot7 |
| value | 4.013 | 8.34 | 12.95 | 26.81 | 61.59 | 82.49 | 71.52 |
| SD | <0.01 | 0.59 | 0.84 | 2.61 | 6.59 | 7.22 | 7.05 |
| Accuracy of LLMs outputs given shots (%) | | | | | | | |
| dataset | shot1 | shot2 | shot3 | shot4 | shot5 | shot6 | shot7 |
| country | 0 | 54.15 | 74.29 | 88.47 | 46.21 | 21.59 | 22.68 |
| animal | 0 | 44.51 | 69.43 | 76.19 | 64.19 | 36.14 | 33.54 |
| color | 0 | 37.49 | 66.51 | 72.18 | 73.16 | 46.95 | 38.49 |
| Accuracy in 4 complex pattern given shots (%) | | | | | | | |
| pattern | shot1 | shot2 | shot3 | shot4 | shot5 | shot6 | shot7 |
| Asia | 0.35 | 3.27 | 4.26 | 15.29 | 34.72 | 84.53 | 79.16 |
| Europe | 3.75 | 8.29 | 11.16 | 24.68 | 49.36 | 89.38 | 89.51 |
| Size | 4.59 | 2.94 | 6.43 | 7.29 | 7.16 | 26.46 | 34.19 |
| Alphabet | 0.11 | 1.26 | 1.47 | 39.16 | 69.17 | 54.91 | 18.67 |

Table 1: The relationship between the accuracy of GPT2-XL outputs and IE by each shot in 3 categories.

## 5.3 Texts Generated from LLMs and Humans Exhibit Different IE Values

We seek to measure the differences in text generated by larger language models compared to human texts, as well as the variations among these LLMs themselves. Specifically, we use questions from OpenHermes as inputs and collect responses by invoking the APIs of GPT-4, Claude3-opus, Claude3-sonnet, and Llama3. These responses were subjected to the same data processing methods described in Section 3.3.2. To evaluate these extremely large and closed-source language models, we implement a 3-step strategy:

---

[7]We randomly sample a total of $N = 1000$ samples and regard the generation to be correct if the generated entity belongs to the corresponding domain of the dataset (country, animal or color) and is not repetitive.

| Text+Estimator | token0 | token1 | token2 | token3 | token4 | token5 | token6 | token7 | token8 |
|---|---|---|---|---|---|---|---|---|---|
| Human+GPT2-XL | 10.9 | 16.9 | 18.6 | 19.5 | 19.5 | 19.7 | 19.6 | 19.5 | 19.4 |
| Human+GEMMA | 9.5 | 16.8 | 22.4 | 24.3 | 24.0 | 25.3 | 24.6 | 25.0 | 25.9 |
| GPT4+GPT2-XL | 11.3 | 18.8 | 23.5 | 27.2 | 34.5 | 37.2 | 39.2 | 39.5 | 39.2 |
| GPT4+GEMMA | 12.1 | 20.5 | 25.1 | 31.6 | 36.3 | 39.9 | 40.4 | 39.5 | 40.6 |
| Claude3-opus+GPT2-XL | 12.6 | 21.8 | 26.6 | 29.5 | 36.8 | 39.8 | 42.6 | 45.2 | 45.3 |
| Claude3-sonnet+GPT2-XL | 11.4 | 17.4 | 24.8 | 28.5 | 32.5 | 36.5 | 36.1 | 36.2 | 36.2 |
| Llama3 (70B)+GPT2-XL | 11.2 | 18.1 | 23.6 | 24.5 | 28.5 | 32.6 | 36.5 | 36.8 | 36.6 |

Table 2: IE in texts generated from human and popular LLMs. "text" refers to the party that generates the text. "Estimator" refers to the LM used to transform the text into representations and estimates the IE value using $f$ described in Section 3.2.

**Step 1:** Collect the answers from these LLMs (or humans) via the questions from the OpenHermes.

**Step 2:** Following the data processing in Section 3.3.2, we format these answers into input sequences of 8 tokens and obtained their representations using smaller LMs (e.g., GPT-2, GEMMA).

**Step 3:** These representations were then processed through an estimator to calculate the mutual information introduced in Section 3.2, thereby determining the IE values of these answers via Equation 7.

Table 2 illustrates an interesting phenomenon: LLM-generated texts exhibited substantially greater IE value than human texts. This observation is intuitive—given that LLMs aim to generate tokens with the highest probability, naturally resulting in greater entropy reduction.

Another observation is that the text generated by different LLMs (GPT-4, Claude3, and Llama3) displays variations in IE values. Significant differences are observed not only in the maximum strength of the IE but also in the patterns of growth. Without actually computing the transformer representations of the target LLMs, these findings open a promising path towards estimating the semantic capturing capability from extremely large and closed-source LMs without expensive computational costs.

## 6 LIMITATIONS

**Position-wise Token**: Given that mutual information intrinsically demands the distribution of two tokens to be valid, we require every token's position to hold a specific meaning, such as representing the beginning or end of a sentence, the subject, predicate, and so forth. Hence, applying our estimator directly to existing tasks may result in a lack of interpretability as the token lengths and positions in the samples vary significantly.

**Sample Amount**: To ensure the accuracy of joint and marginal distributions of high-dimensional continuous representations, a tremendous number of samples is essential. We are attempting more mechanistic alternative methods, hoping to reduce sample size requirements in the future.

**More Models and Tokens**: It is evident that our experiments lack larger-sized models and analysis of long-length texts, especially for emergence and hallucination analysis. Given more computational resources, we would continue to expand these experiments.

## 7 CONCLUSION

In this paper, we mathematically model the entropy of tokens and propose a quantitative metric, Information Emergence (IE), representing the LLM's ability to obtain semantics from tokens. Under the proposed low-resource estimator, we corroborate that IE possesses semantic faithfulness and sensitivity not found in other metrics. under the settings of ICL and natural sentences, We conducted extensive experiments explaining why ICL provides clearer semantics than natural text, as well as the intrinsic relationship between IE and hallucinations. Simultaneously, we discovered that IE can be utilized to distinguish whether the source of the text originates from humans or LLMs, particularly a simple and feasible strategy for those of significantly large LMs.

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

# A  ALGORITHM FOR ESTIMATING MUTUAL INFORMATION

---

**Algorithm 1** Estimating Mutual Information

---

**Require:** : A set of input tokens $U \in \mathbb{R}^{S*T}$, where $S$ denotes the total number of samples and $T$ represents the number of token in each sample. A LLM $f_\tau$ with $L$ layer of blocks and hidden state dimension $D$. A estimator $f_\theta$.
**Ensure:** : Mutual Information $M \in \mathbb{R}^{L*T}$.
  **procedure 1** Extracting Representation $H \in \mathbb{R}^{S*L*T*D}$ from LLM
  Initialization: $H = \emptyset$
  **for** each sample $s$ in $S$ **do**
    $H \leftarrow H + f_\tau(U_s)$
  **end for**
  **procedure 2** Estimating Mutual Information $M$
  Initialization: $M = \emptyset, l = 0, t = 0,$
  **while** $l < L$ and $t < T$ **do**
    $I_x \leftarrow H_{l,t}(H_{l,t} \in \mathbb{R}^{S*D})$
    $I_y \leftarrow H_{l+1,t}(H_{l+1,t} \in \mathbb{R}^{S*D})$
    Shuffle $H_{l+1,t}$ in the dimension $S$
    $I_y* \leftarrow H_{l+1,t}(H_{l+1,t} \in \mathbb{R}^{S*D})$
    $input1 \leftarrow I_x || I_y$
    $input2 \leftarrow I_x || I_y*$
    Initialization: $M_{tmp} = 0$
    **for** Epoch $i < 10k$ **do**
      $output1 \leftarrow f_\theta(input1)$
      $output2 \leftarrow f_\theta(input2)$
      $\mathcal{L} = \frac{1}{S}\sum_{s=1}^{S}(output1) - \log(\frac{1}{S}\sum_{s=1}^{S}(output2))$
      $\mathcal{L}$ backpropagation
      **if** $M_{tmp} == 0$ **then**
        $M_{tmp} \leftarrow -log_2^e\mathcal{L}$
      **else if** $M_{tmp} \neq 0$ **then**
        **if** $M_{tmp} < -log_2^e\mathcal{L}$ **then**
          $M_{tmp} \leftarrow -log_2^e\mathcal{L}$
        **end if**
      **end if**
    **end for**
    $l \leftarrow l + 1, t \leftarrow t + 1$
    $M_{l,t} \leftarrow M_{tmp}$
  **end while**
  **return** $M$

---

Algorithm 1 is employed to elucidate the entire process of estimating mutual information. Simplified, the method involves two primary steps: Step 1 involves extracting representative samples from a LLM, and Step 2 entails estimating the mutual information between these representation samples. We denote the time required to estimate a pair of representations ($H_{l,t}$ and $H_{l+1,t}$) as $\alpha$. Consequently, the time complexity for estimating representations from an LLM for a sequence $S_T = token1, token2, \ddot{,}tokenT - 1$ across $L$ block layers is denoted as $O(LT\alpha)$.

In practical implementations, $\alpha$ approximately costs 40 minutes on one 3090 GPU, whereas significant improvements on a 4090 GPU reduce this time to about 20 minutes.

# B  CASES IN ARITHMETIC TASKS

We have selected a total of 8 arithmetic tasks, as illustrated in the caption of Figure 3. For these tasks, we employed the 2-shots as the prompt templates for the ICl method. We randomly matched different shots for each sample. A representative example from each task is selected and presented as follows:

**1 digit addition:**

**OpenOrca**

|  | token0 | token1 | token2 | ⋯⋯ | token-3 | token-2 | token-1 |
|---|---|---|---|---|---|---|---|
| with previous token | 8.38 | 13.42 | 15.56 | | 17.11 | 17.16 | 17.26 |
| wo. previous token | 8.38 | 10.75 | 10.63 | | 10.69 | 10.64 | 12.85 |

**OpenHermes**

|  | token0 | token1 | token2 | ⋯⋯ | token-3 | token-2 | token-1 |
|---|---|---|---|---|---|---|---|
| with previous token | 7.91 | 12.37 | 15.03 | | 18.37 | 18.43 | 19.22 |
| wo. previous token | 7.91 | 10.78 | 10.91 | | 10.88 | 10.95 | 12.62 |

(a) GPT2-XL

**OpenOrca**

|  | token0 | token1 | token2 | ⋯⋯ | token-3 | token-2 | token-1 |
|---|---|---|---|---|---|---|---|
| with previous token | 10.53 | 17.27 | 20.17 | | 21.27 | 21.66 | 22.41 |
| wo. previous token | 10.53 | 10.65 | 10.49 | | 10.58 | 10.48 | 10.62 |

**OpenHermes**

|  | token0 | token1 | token2 | ⋯⋯ | token-3 | token-2 | token-1 |
|---|---|---|---|---|---|---|---|
| with previous token | 9.59 | 16.80 | 22.28 | | 25.31 | 25.33 | 25.54 |
| wo. previous token | 9.59 | 9.66 | 9.62 | | 9.48 | 9.67 | 9.61 |

(b) GEMMA

Figure 5: Mutual information of each token position in two datasets, taking GPT2-XL and GENNA as examples.

| measure | t0 | t1 | t2 | t3 | t4 | t5 | t6 | t7 |
|---|---|---|---|---|---|---|---|---|
| baseline | 4.69 | 4.69 | 9.46 | 9.37 | 15.32 | 15.02 | 28.44 | 29.47 |
| model1 | 4.64 | 4.69 | 9.44 | 9.45 | 15.09 | 14.67 | 27.59 | 28.67 |
| model2 | 4.64 | 4.68 | 9.44 | 9.28 | 15.27 | 14.66 | 44.08 | 36.37 |
| model3 | 4.69 | 4.68 | 9.47 | 9.45 | 15.29 | 15.54 | 52.28 | 85.61 |
| token1 | 2.82 | 2.83 | 6.88 | 6.85 | 11.08 | 10.95 | 16.83 | 16.09 |
| token2 | 3.60 | 3.60 | 7.36 | 7.29 | 11.15 | 11.08 | 15.86 | 14.96 |
| candidate | 3.26 | 3.26 | 7.22 | 7.22 | 11.44 | 11.35 | 17.15 | 17.33 |
| fusion1 | 3.84 | 3.84 | 7..63 | 7.64 | 11.66 | 11.49 | 17.45 | 16.28 |
| fusion2 | 3.45 | 3.45 | 7.26 | 7.05 | 11.05 | 11.06 | 16.45 | 16.59 |
| space | 4.69 | 4.69 | 9.46 | 9.37 | 15.32 | 15.02 | 22.44 | 5.19 |
| prefix | 4.69 | 4.69 | 9.46 | 9.46 | 15.32 | 15.34 | 37.85 | 41.05 |

Table 3: "Ablation Study" of how IE value changes with different measures adopted. t0-t7 represent 1st - 8th token.

*"What is 1 plus 0? A: 1, What is 4 plus 4? A: 8, What is 2 plus 7? A:"*

**1 digit division:**

*"What is 6 divided by 1? A: 6, What is 8 divided by 4? A: 2, What is 3 divided by 3? A:"*

**1 digit multiplication:**

*"What is 1 times 8? A: 8, What is 5 times 0? A: 0, What is 6 times 7? A:"*

**1 digit subtraction:**

*"What is 5 minus 2? A: 3, What is 7 minus 6? A: 1, What is 9 minus 0? A:"*

**2 digit addition:**

*"What is 53 plus 97? A: 150, What is 89 plus 25? A: 114, What is 75 plus 63? A:"*

**2 digit division:**

*"What is 72 divided by 9? A: 8, What is 81 divided by 27? A: 3, What is 18 divided by 3? A:"*

**2 digit multiplication:**

*"What is 95 times 55? A: 5225, What is 92 times 88? A: 8096, What is 43 times 42? A:"*

**2 digit subtraction:**

*"What is 25 minus 14? A: 11, What is 55 minus 36? A: 19, What is 80 minus 38? A:"*

## C  MUTUAL INFORMATION IN THE NATURAL SCENARIO

We observed variations in the IE statistics of tokens at different positions within a sentence. Consequently, we systematically evaluated tokens at various positions within a sentence, as illustrated in Figure 5. Specifically, token0, token1, and token2 were derived from the same sample set A from OpenOrca, while token-3, token-2, and token-1 were taken from another sample set B from OpenOrca. Sample set A ensured that token0 was the initial token of the sentence, while set B ensured that token-1 was the last token of the sentence. This allowed us to measure differences in IE statistics for tokens at the beginning, middle, and end of sentences across variable sentence lengths.

Figure 5 presents an interesting phenomenon: taking GPT2-XL and GEMMA as examples, GPT2-XL exhibits distinct responses to tokens at different sentence positions—IE values increase at the beginning, stabilize in the middle, and rise again at the end. GEMMA, on the other hand, does not display such positional sensitivity. We hypothesize that this may be related to the different preprocessing methods used in the training data.

| Model | categories | shot3 | shot4 | shot5 | shot6 | shot7 |
|---|---|---|---|---|---|---|
| | country | ↑5.67 | ↑12.95 | ↑2.75 | ↑0.33 | ↑1.04 |
| GPT2-large | animal | ↑4.24 | ↑6.06 | ↑9.52 | ↑0.39 | ↓1.44 |
| | color | ↑4.88 | ↑4.82 | ↑6.39 | ↑1.24 | ↑0.22 |
| | country | ↑5.86 | ↑13.12 | ↑3.56 | ↑1.75 | ↑0.32 |
| GPT2-XL | animal | ↑4.21 | ↑5.75 | ↑8.21 | ↑1.15 | ↑0.61 |
| | color | ↑3.82 | ↑4.61 | ↑7.06 | ↑1.74 | ↑0.54 |
| | country | ↑6.33 | ↑22.16 | ↓2.86 | ↑3.21 | ↓3.54 |
| Gemma | animal | ↑4.09 | ↑6.24 | ↑8.45 | ↑36.51 | ↓2.14 |
| | color | ↑4.65 | ↑5.16 | ↑7.81 | ↑16.49 | ↑1.21 |
| | country | ↑6.33 | ↑45.26 | ↑7.54 | ↑4.65 | ↓3.15 |
| OpenLlama | animal | ↑4.95 | ↑7.54 | ↑35.16 | ↑2.16 | ↑3.26 |
| | color | ↑4.39 | ↑5.27 | ↑27.56 | ↑11.42 | ↑2.51 |

Table 4: $\Delta \widehat{E}(t)$ compared to the previous token. The red represents $\widehat{E}(t)$ decreases compared to the previous token.

## D    ABLATION STUDY FOR SEMANTIC SENSITIVITY

To investigate the influence of different factors on IE value, we treat the performance of GPT2-XL on the "country" dataset as a baseline and implemented a series of variations. First, we replace GPT2-XL with other LMs, namely **model1** using GPT2-large, **model2** using GEMMA, and **model3** using OpenLlama. Second, we vary the dataset, forming **data1** using "animal" dataset and **data2** using "color" dataset. In addition, we use **candidate** to denote reduced candidates in the original "country" dataset (reducing the total number of countries from 25 to 15), and **fusion1, fusion2** to denote mixed candidates where fusion 1 mixes data from "country" and "animal" domains, and fusion 2 mixes data from "country", "animal", and "color" domains. Last, we alter the input sequence, forming **space** by replacing the *entity* in the 4th shot with *space+entity*[8]. **prefix** prepends a *comma* to the original first token.

As shown in Table 3, Differences in model IE become apparent only when a sufficient number of shots are provided. Statistically, models with larger parameter sizes exhibit higher IE. However, differences in data are apparent starting from the first shot, likely related to the domain of training data (token1, token2). Furthermore, depleting the diversity of shots effectively reduces the IE values in ICL (candidate). Lastly, the format of the prompt significantly influences IE, explaining LLMs' sensitivity to certain perturbations (space, prefix).

## E    SUPPLEMENTARY MATERIALS FOR FINDING 1

### E.1    LIMIT NUMBER OF SHOTS

In Section 3.3.1, we expanded the 3 categories into input sequences containing 10 shots (20 tokens each). Table 4 illustrates the changes in IE value for each shot relative to its predecessor within these sequences. The IE value of 4 different LLMs generally approached their upper limits by the 6th and 7th shots. It is important to note that these results only indicate the existence of an upper limit to the contribution of shot quantity to IE in ICL. They do not imply that the 6th and 7th shot universally represents the upper limit for all ICL tasks.

### E.2    SHOT LENGTH

To examine the IE value associated with shot lengths, we designed a shot format pertinent to sentiment analysis as follows:

*"[entity] sentiment: [label],"*

where *"[entity]"* represents emotional words such as *"happy,"*, *"thrill"*, *"offended"*, etc., and *"[label]"* options include *"positive"* or *"negative,"* specifically chosen based on the category of

---

[8]In the GPT-2 tokenizer, *space+entity* is treated as a new token.

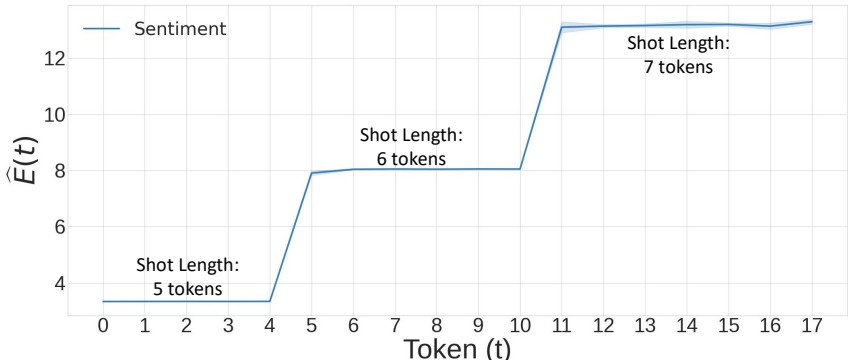

Figure 6: $\widehat{E}(t)$ with inputs of $18$ tokens, consisting of 3 shots in $5$ tokens, $6$ tokens, $7$ tokens, respectively.

*"[entity]"*. The token length of *"[entity]"* was employed to control the overall length of the shot; for instance, when *"[entity]"* consists of single-token words like *"anger," "love,"* etc., the entire shot spans 5 tokens, whereas for two-token words like *"hopeful," "resentful,"* etc., the shot extends to 6 tokens. Consequently, we generated 300,000 input samples, each 18 tokens in length, comprising 3 shots with lengths of 5 tokens, 6 tokens, and 7 tokens respectively.

Figure 6 corroborates our hypothesis: within each shot, all tokens share a uniform IE value. This observation supports another intuitive viewpoint of ICL: an LLM gains greater confidence in the correctness of its predictions only when a new shot is introduced.

### E.3 CASES OF INACCURATE GENERATIONS WITH EXCESSIVE SHOTS

In Table 1 we found 2 types of erroneous repetition, we listed some cases of them from GPT2-XL, in which blue text indicates the shots as prompt, the green text indicates correct entities and red text indicates wrong entities:

**Case 1**: The sequence breakdown was precipitated by the output of an incorrect entity.

*"Ukraine, Mexico, Russia, Australia, New Zealand, United Kingdom, United States, Canada, United States of America, United States of America, United States of America, United States of America"*

**Case 2**: Due to a loop spaning the shots, no new entities were generated.

*"Canada, France, Turkey, Iran, Russia, Ukraine, United Kingdom, United States, Canada, Germany, United States, Canada, Germany, United States, Canada, Germany, United States, Canada, Germany"*

## F DETAILED MUTUAL INFORMATION TABLES

Tables 5 and 6 present the performance of GPT2-XL on the Animal category and OpenOrca datasets, respectively. Although "shots" and "natural sentences" demonstrate different patterns, they share a common characteristic: mutual information increases with token length, aligning well with the NTP mechanism.

| layer | token0 | token1 | token2 | token3 | token4 | token5 | token6 | token7 |
|-------|--------|--------|--------|--------|--------|--------|--------|--------|
| 1 | 2.83 | 2.83 | 6.89 | 6.50 | 10.68 | 9.24 | 14.16 | 11.53 |
| 2 | 2.83 | 2.83 | 6.90 | 6.91 | 11.08 | 11.10 | 16.70 | 16.79 |
| 3 | 2.83 | 2.83 | 6.89 | 6.88 | 11.17 | 11.17 | 16.93 | 16.00 |
| 4 | 2.83 | 2.83 | 6.89 | 6.88 | 11.08 | 11.06 | 16.74 | 16.58 |
| 5 | 2.83 | 2.83 | 6.89 | 6.89 | 11.13 | 11.11 | 16.94 | 15.88 |
| 6 | 2.83 | 2.83 | 6.88 | 6.89 | 11.16 | 11.16 | 18.89 | 17.11 |
| 7 | 2.84 | 2.83 | 6.89 | 6.88 | 11.15 | 11.14 | 16.97 | 17.08 |
| 8 | 2.83 | 2.83 | 6.88 | 6.88 | 11.12 | 11.19 | 16.86 | 17.42 |
| 9 | 2.83 | 2.83 | 6.90 | 6.88 | 11.20 | 11.17 | 16.92 | 15.97 |
| 10 | 2.83 | 2.83 | 6.88 | 6.88 | 11.08 | 11.14 | 16.97 | 16.51 |
| 11 | 2.83 | 2.83 | 6.88 | 6.88 | 11.04 | 11.05 | 17.05 | 16.19 |

Table 5: Mutual information of GPT2-XL in Animal category. Red represents the highest value in this block.

| layer | token0 | token1 | token2 | token3 | token4 | token5 | token6 | token7 |
|-------|--------|--------|--------|--------|--------|--------|--------|--------|
| 1 | 8.40 | 13.71 | 16.01 | 17.33 | 17.21 | 17.67 | 17.95 | 17.29 |
| 2 | 8.36 | 13.77 | 15.76 | 17.06 | 17.08 | 17.72 | 17.68 | 18.00 |
| 3 | 8.44 | 13.75 | 16.09 | 17.10 | 17.82 | 17.69 | 17.84 | 18.04 |
| 4 | 8.44 | 14.20 | 16.07 | 17.29 | 17.45 | 17.74 | 18.51 | 17.81 |
| 5 | 8.39 | 13.50 | 16.32 | 16.83 | 17.82 | 17.98 | 18.26 | 18.14 |
| 6 | 8.41 | 13.69 | 16.03 | 16.99 | 17.58 | 17.82 | 17.52 | 18.33 |
| 7 | 8.41 | 13.68 | 16.06 | 17.00 | 18.32 | 17.72 | 17.69 | 18.19 |
| 8 | 8.40 | 13.80 | 15.97 | 17.26 | 17.61 | 17.73 | 17.52 | 18.44 |
| 9 | 8.35 | 13.69 | 15.95 | 17.17 | 17.21 | 17.54 | 17.47 | 18.03 |
| 10 | 8.41 | 13.57 | 16.30 | 16.57 | 17.46 | 17.85 | 17.85 | 17.51 |
| 11 | 8.34 | 13.37 | 16.02 | 15.93 | 17.30 | 17.24 | 17.27 | 16.91 |

Table 6: Mutual information of GPT2-XL in OpenOrca dataset. Red represents the highest value in this block.

