# OpenReview forum: "On the Entropy of Language Models in Getting Semantic from Tokens"
_ICLR.cc/2025/Conference — ICLR 2025 Conference Withdrawn Submission_

### Official Review · Reviewer_w6Ud · 2024-10-27

**Soundness:** 3
**Presentation:** 2
**Contribution:** 2
**Rating:** 5
**Confidence:** 2

**Summary:**

To quantify the semantic understanding capability of LLMs, this paper introduces a novel metric called Information Emergence, IE for short . IE is defined as the difference in entropy reduction between individual tokens and entire sequences representations within transformer models. Authors propose a mathematical formalism and a practical estimation algorithm to compute IE, which is validated through comprehensive experiments across various scenarios. The paper demonstrates that IE correlates with specific hallucination behaviors and can distinguish between human-written and model-generated texts.

**Strengths:**

1. The concept of Information Emergence (IE) is novel and provides a fresh perspective on evaluating the semantic understanding of LLMs. 2. The paper is methodologically rigorous, with a clear mathematical formulation of IE and a practical estimation algorithm based on mutual information.
3. The proposed IE metric has broad applications, including detecting hallucinations and distinguishing between human and LLM-generated texts.

**Weaknesses:**

1. The motivation is not clear, which is my main concern. I'm not convinced by the introduction on why we need a metric to quantify the behavior of finer-grained tokens, and why other methods fails.
2. Need more discussion on related works and baseline comparison. And I hope to see more baseline comparisons (even designing a straightforward metric or adapting some other approaches for this problem).
3. While the experiments are comprehensive, the paper could benefit from a broader range of model sizes and types. E.g., at least a model >=7B.
4. The method need for a large number of samples to ensure the accuracy of joint and marginal distributions.

**Questions:**

Could you please provide some straightforward baselines and compare them with your metric?

---

### Official Review · Reviewer_pW7s · 2024-11-04

**Soundness:** 1
**Presentation:** 1
**Contribution:** 1
**Rating:** 1
**Confidence:** 3

**Summary:**

This paper proposed a new metric for measuring the capability of modeling semantics of LLMs. The metric concerns the reduction of entropy if conditioned on a longer sequence than a single token. The authors consider the process of generation in LLMs a Markov chains, and measures the mutual information of token embeddings between different layers.

**Strengths:**

N/A

**Weaknesses:**

- The argument of the paper is hard to understand. The authors claimed to mathematically model the entropy of the tokens, but the log-probability of each token and the perplexity metric exactly contains the entropy of the tokens. I do not understand the argument behind the authors' proposal.
 - The authors did not properly introduce their notion of "semantics". In prior research, semantics can be understood under denotational, operational, or distributional contexts. It should be discussed under which setting the author is situating their research.
 - The token embedding of layer $(i+1)$ given layer $i$ is a deterministic process, but the authors cast this as a Markov stochastic process. Please elaborate on why this is stochastic.

**Questions:**

- L077: Consider revising these bibtex entries: GEMMA (Team et al., 2024). This clearly should be "(GEMMA Team, 2024)".
 - L082: Please elaborate what is "ICL-style texts" here.
 - L151: With embeddings as inputs and outputs, a Transformer is deterministic, thus not even stochastic. Please explain.
 - L160: It is not straightforward to sample sequences from BERT et al., so technically these are not language models (defined as distribution over sequences).
 - L208: so $h_l^{\rm ma}$ is the last token?
 - Figure 2: "Increasement" => "increase".
 - "Texts generated from LLMs vs humans": Since you are eliciting *new* responses from humans, these are unseen data for LLMs. The phenomena you observed may not be true if you test LLM with human-generated text in the training set of LLMs.
 - "LLM-generated text exhibited greater IE than human text": Clearly, LLM-generated text exhibits lower perplexity than newly elicited human text. This experiment does not show that IE is superior than existing measures.

---

### Official Review · Reviewer_hwna · 2024-11-04

**Soundness:** 1
**Presentation:** 1
**Contribution:** 2
**Rating:** 3
**Confidence:** 3

**Summary:**

This paper presents a method for determining “information emergence” (IE) that is roughly focused on entropy reduction upon observing a new token. This is introduced as an idea where a model which better understands the semantics of tokens should  have higher entropy reduction than smaller models. This can be estimated by calculating the difference between mutual information between the token hidden state distributions between layers for the full sequence and for individual tokens. Using the framing of IE, the authors describe its implementation for LLMs and follow-up findings on GPT-2, GEMMA, and OpenLlama, like the correlation between IE, model size, and accuracy.

**Strengths:**

1. The paper proposes a formalism of “macro” and “micro” variables to describe the notion of IE in LLMs. This formalism corresponds with their supervenience hypothesis, which in turn is inspired by the term “emergence” from philosophy/information theory.

2. In the experiments, the calculated IE value appears to correlate positively with model size and task accuracy.

**Weaknesses:**

1. The experimental methodology is not explained in a way I can follow. I do not understand the connection between Section 3.3 and Section 4.3. Semantic faithfulness is not defined in this paper or included as a citation, nor is semantic sensitivity. Big-Bench was not mentioned as a dataset until Section 4, and seems to no longer be affected by the token position problem?

2. The findings in the introduction are not strongly supported by the experiments with sufficient evidence. Of the 3 main interesting findings:

a.) Finding 1: Sec 5.1, is somewhat supported, that IE increases token-by-token in natural texts, but the ICL setting is too contrived because the example itself is only a single token, and the other token (comma) is not information-heavy. A equally plausible explanation is that IE does not increase on punctuation (which is semantically less meaningful).;

b), Finding 2: Sec 5.2  Is it surprising that the variance increases when IE increases? To me, this suggests that IE is invariant to scaling, and not that it corresponds to hallucination. IE actually goes down as the number of shots increases, and so I expect the SD is going down too. Even looking closely at Fig 4, it is not clear to me that SD is increasing as shots increases (e.g. 4(d)) has low SDs throughout, 4(a) has SD only increase at the 5th/6th/7th token, but it looks relatively stable as a fraction of E(t) for 4(c)

c) Finding 3: Sec 5.3 This result is interesting as a means for detecting LLM-generated and human-generated text, and perhaps can be the main focus of the 3 findings. However the results are still a bit limited because the methodology is unclear. Questions like how many samples (questions) were taken from OpenHermes? How were these answers formatted into 8 tokens (what if they were too long)? How were the LLMs prompted to answer the questions?


3. I’m confused why there needs to be a distinction between “emergence” as defined in this paper from “emergent abilities” as used for LLMs, and neither this paper nor the cited papers appear to explain this. So, what precisely, is the definition of emergence and LLM emergence, and why are they different?

Some prominent citations (like Wei et al., 2022) are not used correctly while 3 of the 4 main citations consistently used to support claims about emergence, including how it is defined do not make strong claims about emergence in models (Liu et al., 2024 – about prompts for power emergency plans; Yu & Dong 2022 – about emergence of complex language learning in L2 (human) students; and Srivastava et al., 2022 – the dataset paper for BIG-Bench. *Note: this is based on reading the abstract as some of these papers are not open-access*). Many of the other citations in the paper, however, look reasonable.

4. As acknowledged by the paper, the limitation of the analysis to specific sequences of equal length and similar token positions is a major obstacle to for using IE at all. Still, it should be possible to create figures like Fig.4 which goes from 0... max_seq_length; why can that not be done in this work?

**Questions:**

1. What is the motivation for the estimation function in Section 3.2? Since the experiments primarily use open-weight models, we have access to the hidden states. These are discrete distributions and we can compute/approximate KL divergence more precisely than learning a separate estimator, f. And a related question is why we should measure the IE at each layer of the Transformer instead of only the final layer(s).

2. I’m also confused by the batch size choice. Is the batch size chosen to be the full size of the data, or is it 300K? Or am I missing something about number of samples vs. batch size?

3. For ICL, the comma is always at even positions for every example in the dataset. Then wouldn’t the macro-level MI (first term) already be small or near 0; the 2nd term will also be 0, and so the IE for that token would be basically 0. Is this what is actually happening, or is the macro MI nonzero?

4. And maybe I’m misunderstanding the micro variables: when calculating the micro variables, are padding tokens used? For example, in L220, for $h_{l}^{mi_1}$, is there a padding/zero token before the word ‘language’ or not?

5. I assume by $l$ and *block*, this is referring to layers of the transformer. The word "layer" was never used, so please correct me if this is the wrong understanding.

Minor edits suggestions:

In Definition 1: Define $h_{l}^{mi\_t}$ earlier -- it isn't defined until the next page.

L227: “Notably, We” -> “Notably, we”

L364: “Moreover, We” -> “Moreover, we”

L242: “how confidence” -> “the confidence”

L753: “ICl” -> “ICL”

Fig 3: “divise” -> “division” or “divide”

The title, and in general, the use of the word "semantic" (adjective) and "semantics" (noun) needs to be more careful. It should probably be "semantics" in the title.

---

> ### Author Response · Authors · 2024-11-18
> **response**
>
> Thank you for your meticulous review and for raising some invaluable suggestions and questions. In response to your rigorous approach, we will provide some simple answers to your questions.
>
> 1. We do not understand your point. Our estimator f is indeed computed through the distribution of the representation. To our knowledge, this is the simplest computational method when mathematical derivations can be verified. You may have misunderstood the motivation behind our estimator.
>
> 2. The 300k refers to the batch size. We clarified in Section 3.3 that the number of samples needs to be above 300k to ensure the stability of distribution sampling, hence our choice of the minimum batch size of 300k.
>
> 3. No, although the representations on the tokens are all commas, the representations will differ due to different contexts. Moreover, if you suspect that the IE does not increase due to the same token, you can find the answer in Figure 6. In Figure 6, not all tokens in each shot are identical, but the conclusion of "stability within the shot" still holds.
>
> 4. No, the micro-variables are always treated as the first token, otherwise, it would not satisfy the emergence theory.
>
> 5. Yes, we have chosen the concept of a block only at the code level.

---

### Note · Authors · 2024-11-18

**Comment:**

Regrettably, after reviewing the comments from the reviewers, we have decided to withdraw our manuscript.

Firstly, we would like to express our gratitude to the reviewer hwna, as we can perceive your meticulous attitude from your comments. However, from the reviewers' feedback, we still sense that the three reviewers did not understand our manuscript sufficiently (as indicated by their confidence levels, it seems the reviewers also acknowledge this). We speculate that the heavy load of review assignments may have prevented the reviewers from dedicating enough time to our manuscript. Therefore, we would like to provide a simple clarification of our manuscript:

This paper revolves around the quantification of semantics and the ability of language models to capture semantics. By integrating knowledge from information theory and utilizing entropy reduction, we describe how semantics gradually become certain during inference. Based on such a mathematical model that allows for the quantification of semantics, we developed a derivable estimator and discovered some patterns of language models from a semantic perspective. We believe our manuscript is theoretically provable, experimentally solid, and the conclusions insightful.

Thank you to all those who have shown interest in reading our work.

**Withdrawal Confirmation:**

I have read and agree with the venue's withdrawal policy on behalf of myself and my co-authors.